# Qualitative Methodology in Translational Health Research: Current Practices and Future Directions

**DOI:** 10.3390/healthcare11192665

**Published:** 2023-10-01

**Authors:** Kritika Rana, Prakash Poudel, Ritesh Chimoriya

**Affiliations:** 1Translational Health Research Institute, Western Sydney University, Campbelltown, NSW 2560, Australia; 2Philanthropy Nepal (Paropakari Nepal) Research Collaboration, Auburn, NSW 2144, Australia; r.chimoriya@westernsydney.edu.au; 3Office of Research and Education, Canberra Health Services, ACT Government, Garran, ACT 2606, Australia; prakash.poudel@act.gov.au; 4School of Medicine, Western Sydney University, Campbelltown, NSW 2560, Australia; 5Concord Institute of Academic Surgery, Concord Repatriation General Hospital, Concord, NSW 2139, Australia

**Keywords:** translational health, qualitative research, qualitative methodology, health equity, health outcomes, sampling, data collection, data analysis

## Abstract

Translational health research is an interdisciplinary field aimed at bridging the gap between basic science studies, preventative studies, and clinical practice to improve health-related outcomes. Qualitative research methods provide a unique perspective on the emotional, social, cultural, and contextual factors that influence health and healthcare and thus are recognized as valuable tools for translational health research. This approach can be embedded within a mixed method design which complements the quantitative findings. This methodological paper aims to provide a comprehensive review of the fundamental concepts and methodologies used in qualitative research, emphasizing their utilization and significance in translational health research. Several approaches to qualitative research methodology are discussed in this review, including ethnography, phenomenology, grounded theory, case study, and action research. Theoretical frameworks such as the social-ecological model, intersectionality, and participatory action research are also examined to provide a structure for understanding and interpreting complex health issues. This methodological paper also reviews commonly used sampling techniques such as purposive, snowball, convenience, theoretical, and maximum variation sampling, along with data collection methods such as in-depth interviews, focus groups, observation, document analysis, and participatory methods. Moreover, data analysis techniques such as thematic analysis, grounded theory, content analysis, narrative analysis, and reflexive analysis, are discussed in the context of translational health. Overall, this review highlights the challenges and opportunities of using qualitative methods in current practice, while also discussing future directions and providing valuable guidance and insights to researchers interested in conducting qualitative research in translational health.

## 1. Introduction

Translational health research is a complex and interdisciplinary field that aims to bridge the gap between preclinical studies and clinical practice, with the goal of improving health-related outcomes [1]. Qualitative research methods have become increasingly recognized as valuable tools for translational health research, as they offer a unique perspective on the social, cultural, and contextual factors that influence health and healthcare [2,3]. As opposed to quantitative methods which rely on numerical data, qualitative methods involve the collection and analysis of non-numerical qualitative data, such as interviews, observations, and documents. This approach allows researchers to explore the experiences and perspectives of patients, healthcare providers, and other stakeholders, and to generate new insights into the complex and dynamic processes that shape health and healthcare [4,5].

Qualitative studies in healthcare translation play a crucial role in enhancing patient care by unraveling the intricate world of emotions [6]. The understanding of the role of emotions in healthcare is in an early state; thus, qualitative research can be utilized in exploring strategies for mitigating safety risks and shifting cultural norms in medicine [7]. Similarly, this can be used in understanding the complex emotional dynamics of patients’ and caregivers’ relationship with healthcare professionals, and how this impacts the management of illness and overall disease trajectory [8]. This further empowers healthcare providers to offer culturally sensitive, empathetic, and patient-centered care which ultimately improves patient experiences and outcomes [6,9]. Similarly, qualitative research is also a valuable tool that can be used in healthcare education, particularly in incorporating emotional intelligence in healthcare education [7]. For instance, recent qualitative research underscores the value of employing emotionally intelligent behaviors in healthcare settings to effectively manage stress and foster better professional relationships among healthcare students and staff [10].

In recent times, the significance of qualitative design has become increasingly apparent, particularly in understanding the intersectionality and aid in the digital transformation within the healthcare sector [11,12]. Since qualitative research has been extensively used in informing the development of quantitative instruments, this design also enhances the understanding of results from quantitative analysis [13]. Qualitative research has been increasingly used in translation health research to better understand user’s expectations and enhance inclusive engagement, thereby developing the translation or implementation process [13,14,15,16]. Similarly, qualitative studies may complement quantitative studies and can be seamlessly included within a larger study design. This methodological paper provides a comprehensive review of the fundamental concepts and methodologies used in qualitative research, emphasizing their significance in translational health research. As a summary of the application of these concepts, Figure 1 illustrates a comprehensive framework for conducting qualitative research in translational health. Furthermore, this review explores how the qualitative method has been used to address study objectives and research questions in the field of translational health research, and highlights some of the strengths, limitations, challenges, and opportunities associated with using qualitative research methods in translation health research.

## 2. Approaches to Qualitative Research Methodology

Qualitative research methodology in translational health research encompasses a diverse range of techniques that aim to explore and understand the subjective experiences, meanings, and social contexts surrounding health phenomena [17]. These methods provide in-depth insights into individuals’ perspectives, behaviors, and beliefs, which allows to uncover complex and nuanced aspects of health [18,19]. Qualitative research uses several approaches and some of the common approaches used in translational health research are provided in Table 1.

## 3. Theoretical Frameworks for Qualitative Methodology in Translational Health Research

Theoretical frameworks provide a map for qualitative exploration by describing concepts and relationships within a phenomenon [38]. These frameworks can be built inductively or based on the existing theories and literature and can help direct attention to the phenomenon of interest [39]. The utilization of theoretical frameworks can be valuable in qualitative methodology for translational health research, as it provides a structured framework for understanding and interpreting the intricate nature of health issues.

### 3.1. Social-Ecological Model

The social-ecological model is a theoretical framework used to understand the complex interplay between individual, interpersonal, institutional, community, and societal factors that shape human behavior and health outcomes [40]. This model recognizes that individuals are not solely responsible for their health and well-being, and a range of environmental and societal factors play a fundamental role in influencing health outcomes. At the individual level, factors such as genetics, knowledge, attitudes, beliefs, and behaviors influence health outcomes. The interpersonal level encompasses social relationships and social networks that individuals form a part of, including families, friends, and colleagues [41,42]. The institutional level focuses on organizations, policies, and social institutions that shape behavior and influence health outcomes. The community level includes factors associated with the physical and social environment, including access to resources and community norms and values. Finally, the societal level integrates broader social, cultural, economic, and political factors that influence health outcomes, including policies, laws, and cultural norms [41,42]. The social-ecological model emphasizes the interconnectedness of these multiple levels and the importance of addressing health issues through a multi-level approach by considering the broader context in which individuals live, work, and interact [43]. For instance, interventions aimed at reducing the rates of obesity may need to target individual-level factors such as knowledge and behaviors, as well as community-level factors such as access to healthy food options and physical activity opportunities, and societal-level factors such as food industry marketing practices and government policies on nutrition.

### 3.2. Intersectionality

Intersectionality is a valuable theoretical framework for understanding the complex and intersecting social identities that shape individuals’ experiences of health and healthcare [44,45]. In health research, intersectionality can help to identify the unique challenges faced by marginalized individuals and communities and can inform interventions and policies that address these challenges [45,46,47]. For instance, a health researcher may use an intersectional lens to explore how the intersection of race, gender, and socioeconomic status impacts individuals’ access to healthcare. The researcher may conduct interviews with individuals from different racial and socioeconomic backgrounds, asking them about their experiences with healthcare providers and their ability to access medical care. By examining the ways in which multiple identities intersect to shape individuals’ experiences, the researcher can gain a deeper understanding of the unique barriers and challenges faced by marginalized communities. Moreover, intersectionality can help health researchers to identify areas of privilege and power within the healthcare system [46]. For example, a health researcher may examine the ways in which gender and sexuality intersect to create unique challenges for LGBTQ+ individuals seeking healthcare. By identifying areas of privilege and power within the healthcare system, the researcher can develop interventions and policies that promote equity and justice.

### 3.3. Participatory Action Research

Participatory action research is a theoretical framework that emphasizes collaboration between researchers and community members. This approach seeks to empower individuals and community members to identify and address health issues that affect them, rather than imposing solutions from the outside [48]. It involves a cyclical process of reflection, planning, action, and evaluation, where researchers work in partnership with community members by involving them as active participants in all stages of the research [48]. This collaborative approach allows for the development of more culturally responsive and relevant interventions and policies, as community members can provide valuable insights into the unique challenges and needs of their communities [49]. Participatory action research in health research focuses on addressing health disparities, promoting community ownership and action, as well as fostering sustainable solutions to health challenges. For instance, in a project focused on mental health services in a marginalized community, researchers and community members may collaborate to identify barriers and co-design interventions. This inclusive approach may lead to tailored and sustainable improvements, such as the development of community-based support programs and policy advocacy to address the mental health needs specific to the community.

## 4. Sampling Techniques in Qualitative Methodology for Translational Health Research

Qualitative research sampling refers to the process of selecting individuals or cases to be included in the research sample. Qualitative research uses a non-probability sample, as the selected sample does not reflect a list of all possible elements in a full population and make inferences of the findings, but is guided by the principle of seeking information-rich cases or individuals who can contribute diverse perspectives and experiences, allowing for a comprehensive exploration of the phenomenon under investigation [50]. As summarized in Table 2, there are several sampling techniques that are commonly used in qualitative methodology for translational health research. There are strengths of each sampling technique, and it is crucial to carefully select the most appropriate technique based on the research question, population of interest, and available resources [51].

## 5. Data Collection Methods in Qualitative Methodology for Translational Health Research

Data collection plays a crucial role in qualitative methodology as it has a direct impact on the accuracy and dependability of research outcomes, and the quality of data collected can determine the validity of study findings [57]. Qualitative methodology in translational health research encompasses various data collection methods to gather rich and nuanced information, which can also be used in combination to gain a comprehensive understanding of the phenomena under investigation.

### 5.1. In-Depth Interviews

In-depth interviews are one of the most used data collection methods in qualitative methodology for translational health research. This method involves conducting individual interviews with participants to gather detailed and rich data on their experiences, beliefs, and perspectives related to the research question [58]. While in-depth interviews allow researchers to explore complex and sensitive topics, the interviews require extensive planning and training to ensure that the interviewer establishes rapport and creates a safe and supportive environment for the participant [59,60]. In-depth interviews in health research allow researchers to gather rich and detailed data on participants’ experiences and perspectives, which can provide valuable insights into the social and cultural factors that shape health outcomes. Moreover, in-depth interviews can be particularly useful for exploring sensitive topics, such as stigmatized health conditions or experiences of discrimination in healthcare [60]. In-depth interviews in health research generally follow a semi-structured or unstructured format, which allows for flexibility and exploration of unexpected themes or topics [58,59]. The interviewer may begin with a set of open-ended questions and follow up on the responses to explore further. To ensure ethical and respectful treatment of participants, in-depth interviews should be conducted in a private and confidential setting, with informed consent obtained prior to the interview [58,61,62].

### 5.2. Focus Groups

Focus groups are also one of the commonly used data collection methods in qualitative methodology for translational health research. In this data collection method, group interviews are conducted with participants who share similar characteristics or experiences related to the research question [63,64]. A focus group involves a moderator who facilitates a discussion among a small group of participants, with the goal of exploring a range of opinions, experiences, and attitudes related to the health issue or topic of interest [64,65]. Focus groups can be particularly useful for exploring topics that may be difficult to discuss in one-on-one interviews, as the group dynamic can create a more comfortable and supportive environment for participants to share their experiences and perspectives [63]. Focus groups also allow participants to discuss and explain to each other the questions/topics discussed. Furthermore, focus groups can be especially valuable for marginalized or stigmatized populations who may feel more comfortable discussing sensitive health topics in a group setting [65]. While focus groups allow researchers to capture diverse perspectives and generate rich and interactive data, this method may also introduce group dynamics and biases that can affect the quality of the data [63,65].

### 5.3. Observation

Observation is a data collection method that involves observing and documenting participants’ behavior, interactions, and environments related to the research question. Observation can be conducted in natural settings or structured environments, such as clinics or hospitals [66]. Observation allows researchers to gather data on participants’ behavior and experiences in real-life situations and understand the context, which will eventually be helpful in planning subsequent interviews. However, it is essential for the observer to have extensive training to ensure that the observer remains neutral and non-intrusive [67]. Participant observation can be very helpful for translational health research as this would enable access to many people and a wide range of information [68].

### 5.4. Participatory Methods

Participatory methods refer to a set of data collection methods that actively involve individuals or communities as active participants in the research process, such as co-design, co-production, and co-research. Therefore, this is a useful tool in translation health research as participants have the opportunity to engage in the design and development of a new treatment/care and they could facilitate the translation process efficiently. Participatory methods allow participants to actively shape the research question, methods, and findings and can lead to more relevant and impactful research [69]. However, participatory methods may be resource-intensive, and require careful consideration of power dynamics, ethics, as well as sustainability [69].

## 6. Data Analysis Techniques in Qualitative Methodology for Translational Health Research

### 6.1. Thematic Analysis

Thematic analysis is a widely used data analysis technique in qualitative phenomenology design for translational health research. It involves identifying and analyzing common patterns, themes, and categories in the data to generate insights and develop a coherent and comprehensive account of the research question [70]. Thematic analysis is more focused on the deductive approach but can be used in inductive approaches. In deductive approaches, researchers begin with pre-existing themes or categories that have been derived from previous research or theory and subsequently apply them to the data. In contrast, in inductive approaches, researchers allow the themes or categories to emerge from the data itself, without imposing any pre-existing framework [71]. The process of thematic analysis usually involves several steps. The first step is to become familiar with the data by reading and re-reading it several times. The next step is to code the data, which involves identifying and labeling segments of data that relate to the same idea or concept. Codes can be descriptive, such as labeling a segment of data as “patient experience”, or conceptual, such as labeling a segment of data as “loss of autonomy”. Once the data has been coded, the next step is to identify and analyze patterns, themes, and categories that emerge from the data [72]. Themes can be derived from similarities or differences in the codes, or from concepts that are repeated across the data [70,72]. The final step is to develop a coherent and comprehensive account of the research question by organizing the themes and categories into a meaningful framework. Thematic analysis allows researchers to capture the complexity and diversity of the data and generate rich insights that can inform policy, practice, and research [72]. It is a flexible and adaptable technique that can be applied to a wide range of research questions and populations, making it a frequently used data analysis technique in qualitative methodology for translational health research [62,70,72]. The key approach and outcome of the analysis are shown in Table 3.

### 6.2. Grounded Theory

Grounded theory is another data analysis technique often used in qualitative methodology for translational health research. This technique involves developing a theory or framework based on the data collected through a process of constant comparison and iteration [28]. Grounded theory allows researchers to generate new and innovative insights and theories that are grounded in the data, and can lead to the development of new practices, policies, and interventions [28].

### 6.3. Reflexive Analysis

Reflexive analysis is a data analysis technique where researchers reflect on their role and positionality in the research process, allowing researchers to critically evaluate their biases and assumptions and generate insights into how they may have impacted the data collected and analyzed [73].

## 7. Leveraging Computer-Assisted Qualitative Data Analysis Software (CAQDAS) for Systematic Qualitative Insights

The analysis of qualitative data represents a complex process and the multi-faceted nature of qualitative data, often comprising rich narratives, interviews, observations, or textual content, and thus demands a systemic approach to generate meaningful insights. To enhance the comprehensibility and rigor of qualitative research, it is recommended to incorporate thorough and systematic analysis mechanisms into each chosen research strategy [74]. One valuable addition would be the utilization of computer programs and software tools specifically designed to aid in the analysis process. These programs provide researchers a systematic and efficient means to code vast datasets, identify patterns, and derive meaningful interpretations [74,75]. By integrating these digital tools into qualitative research strategies, researchers not only streamline the analytical process but also increase the transparency and replicability of their work. This orientation can serve as a roadmap, enabling interested individuals to explore these analysis strategies in more depth through specialized texts or resources. There are numerous Computer-assisted Qualitative Data Analysis Software (CAQDAS) such as NVivo (QSR International, Burlington, MA, USA), Atlas.ti (Scientific Software Development GmbH, Berlin, Germany), Dedoose (SocioCultural Research Consultants/UCLA, Los Angeles, CA, USA), and QDA Miner (Provalis Research, Montreal, QC, Canada) [74]. NVivo, which offers a variety of packages including NVivo Starter, NVivo Pro, and NVivo Plus, is widely used in qualitative health translational research [75]. Recently, Quirkos (Quirkos, Edinburgh, Scotland, UK) has also been widely used which is facilitated by its graphic interface to understand qualitative data [76,77].

## 8. Ethical Considerations in Qualitative Methodology for Translational Health Research

Ethical considerations in qualitative methodology are crucial to ensure that research is conducted responsibly, respecting the rights and well-being of the participants. Consideration of ethical challenges is particularly important while conducting qualitative research in the field of translational health due to the potential vulnerabilities and sensitive nature of health-related topics [78].

### 8.1. Informed Consent

Informed consent is a fundamental ethical principle in research, which requires that participants provide voluntary, informed, and ongoing consent to participate in a study [79,80]. In qualitative research, obtaining informed consent involves providing participants with information about the study, including its purpose, procedures, potential risks and benefits, and their rights as participants. Participants should be given the opportunity to ask questions and to withdraw from the study at any time. It is important to ensure that participants have the capacity to give informed consent and that they fully understand the implications of their participation.

### 8.2. Confidentiality and Anonymity

Confidentiality and anonymity are crucial ethical considerations in qualitative research [81]. Participants should be assured that their personal information and identities will be kept confidential and that their anonymity will be protected. Researchers must ensure that participant data is kept secure and only accessible to authorized personnel. Moreover, researchers must ensure that the data is stored in a secure location and that any identifying information is removed or disguised in the analysis and dissemination of findings [81].

### 8.3. Power Dynamics

Power dynamics exist in all research relationships, and it is important for researchers to be aware of and sensitive to these dynamics [80]. Researchers must ensure that participants feel comfortable and empowered to share their experiences and perspectives. This may involve creating a safe and supportive research environment, using language and terminology that is culturally and linguistically appropriate, and acknowledging and respecting the diversity of experiences and perspectives among participants.

### 8.4. Beneficence and Non-Maleficence

The principles of beneficence and non-maleficence require that researchers strive to maximize benefits and minimize harm to participants [80]. In qualitative research, this may involve ensuring that participants are not subjected to emotional distress or psychological harm, providing appropriate support and resources for participants who may require assistance, and ensuring that the research findings are used to inform the development of culturally responsive and effective interventions and policies that promote health equity.

### 8.5. Rigor and Trustworthiness of the Research

Rigor is a critical aspect of qualitative inquiry, aiming to establish the credibility, transferability, dependability, and confirmability of the findings [82]. A range of methodological strategies can be employed to ensure the rigor and trustworthiness of the research [82,83]. One key measure would be conducting interviews with a researcher who has undergone specialized training in qualitative research methods, ensuring a high level of proficiency and sensitivity to the nuances of qualitative inquiry [82]. Similarly, debriefing sessions may be helpful to review the completeness of the gathered data and to identify potential areas for further exploration, and these sessions may continue until data saturation is achieved [84]. It is also essential to reach a consensus on coding through collaborative efforts within the research team. The results presented may be fortified by the inclusion of direct quotes from participants, providing a robust foundation for the findings [83]. Moreover, the study design and reporting of results may also be guided by established guidelines, such as the Consolidated Criteria for Reporting Qualitative Studies (COREQ), to ensure comprehensive and transparent reporting of qualitative research [85].

The key components of the qualitative analysis in practice are shown in Figure 2.

## 9. Challenges and Limitations of Qualitative Methodology in Translational Health Research

Qualitative research can provide rich and detailed insights into the complex social and cultural factors that influence health outcomes and can inform the development of more effective and culturally responsive interventions and policies [2,59]. However, qualitative methodology also presents several challenges and limitations that must be considered and addressed to ensure the rigor and validity of the qualitative research [83].

### 9.1. Subjectivity and Bias

One of the primary challenges of qualitative research is the potential for subjectivity and bias in data collection, analysis, and interpretation [86]. Qualitative research relies heavily on the perspectives and experiences of participants and researchers, which can be influenced by a variety of factors, including personal biases, cultural norms and values, and power dynamics [87]. Researchers must be aware of their own biases and take steps to minimize their influence on the research, such as using multiple data sources and methods, engaging in reflexive practice, and using triangulation to verify the study findings.

### 9.2. Sample Size and Generalisability

Another challenge of qualitative research is the relatively small sample sizes that are often used, which can limit the generalizability of the research findings. Qualitative research generally seeks to explore in-depth experiences and perspectives of a specific group or population, rather than seeking to make generalizations to larger populations [88]. However, this can be a challenge in translational health research, where policymakers and healthcare providers may need information that is generalizable to larger populations in order to make informed decisions [89].

### 9.3. Validity and Reliability

Qualitative research involves a complex and iterative process of data collection, analysis, and interpretation, which can be influenced by a variety of factors [88]. Therefore, ensuring the validity and reliability of qualitative research can also be a major challenge. Researchers must take steps to ensure that their findings are valid and reliable, such as using rigorous data collection and analysis methods, engaging in reflexivity and triangulation, and seeking feedback from participants and other stakeholders [90].

### 9.4. Ethical Considerations

The consideration of varied ethical concerns involved in qualitative research in the field of translational health is also a challenge. Qualitative research in translational health involves working closely with participants and engaging in sensitive and potentially emotional topics [88]. Therefore, researchers must ensure that the qualitative research is respectful and culturally appropriate, with strategies in place to minimize the potential for harm or distress to participants [80,91]. Ethical considerations may involve ensuring that participants have provided informed consent, along with maintaining confidentiality and anonymity, and providing appropriate support and resources for participants who may require assistance.

### 9.5. Time and Resource Constraints

Qualitative research can be time-consuming and resource-intensive, which can present challenges for researchers in the field of translational health [92,93]. Qualitative research generally involves a lengthy process of data collection, analysis, and interpretation, which can be further complicated by the need to collaborate with diverse populations and stakeholders [93]. Researchers must ensure that they have adequate time, resources, and support to conduct their research effectively and to address any challenges or limitations that may arise during the study duration.

## 10. Future Directions and Implications of Qualitative Methodology in Translational Health Research

Future directions for qualitative research in the field of translational health include promoting health equity, understanding emotions in healthcare, using participatory research approaches, leveraging digital technology, and incorporating mixed methods approaches [94,95,96]. Evidence suggests that emotions play an integral role in healthcare; however, the area is least explored in terms of how it influences clinical practice [7,96]. Qualitative methodology can be employed to provide insights into the underlying emotions, the social determinants of health, and health disparities. Hence, qualitative research facilitates identifying the underlying causes of health inequities, such as systemic racism or social exclusion, and can help inform interventions to address these issues. In addition, by centering the experiences and perspectives of marginalized communities, qualitative research can help ensure that interventions are culturally sensitive and appropriate [97,98,99]. Participatory research is a collaborative approach that involves engaging community members in all stages of the research process [48] and is an important part of the translation health research. This approach can help ensure that research questions are relevant to the community and that interventions are designed in partnership with community members. Participatory research can also help address power imbalances in the research process and can increase community engagement and ownership of the research findings [48,49]. The comparison of traditional manual and digital data collection and coding in qualitative research has not been extensively investigated [100]. This choice may be contingent on factors such as project scale, available funds, and time, as well as the researcher’s inclination and proficiency in the chosen method. However, digital technology offers new opportunities for qualitative research, such as online focus groups or social media analysis [94,101]. These methods can increase access to hard-to-reach populations and can provide new insights into online communities and social networks [94]. However, researchers must be aware of potential biases and ethical considerations associated with online research [101]. Mixed methods research combines both qualitative and quantitative approaches in a single study. This approach can provide a more comprehensive understanding of complex issues and can allow for triangulation of data [102]. However, mixed methods research requires careful integration of both approaches and may be resource-intensive. Despite the potential benefits of qualitative methodology in translational health research, there are also several challenges that must be addressed. A major challenge is the lack of standardization in qualitative research, which can make it difficult to compare and synthesize findings [103]. There is also a risk of researcher bias in qualitative research, which can impact the validity and reliability of findings. The nuanced perspectives generated from the qualitative research can inform the development of policies and practices that are more responsive and tailored to diverse populations [104]. By incorporating these insights, health systems can strive for greater inclusivity and effectiveness in delivering quality healthcare.

## 11. Conclusions

Qualitative research methodology in translational health research provides a valuable lens to explore the social and cultural factors that shape health and healthcare, thus informing the development of interventions and policies that better address the health needs of marginalized communities. However, researchers must carefully consider the limitations and challenges associated with qualitative research and take steps to address these through rigorous research design and methodology. By embracing participatory approaches and leveraging digital technology, qualitative research in the field of translational health can play a critical role in advancing health equity and ensuring that interventions are culturally responsive and effective for diverse populations. To truly enhance translational health research and practice, we must invest in further studies, prioritize education, and advocate for policy changes that fortify the rigor and impact of qualitative methodologies in shaping a more inclusive and effective healthcare landscape.

## Figures and Tables

**Figure 1 healthcare-11-02665-f001:**
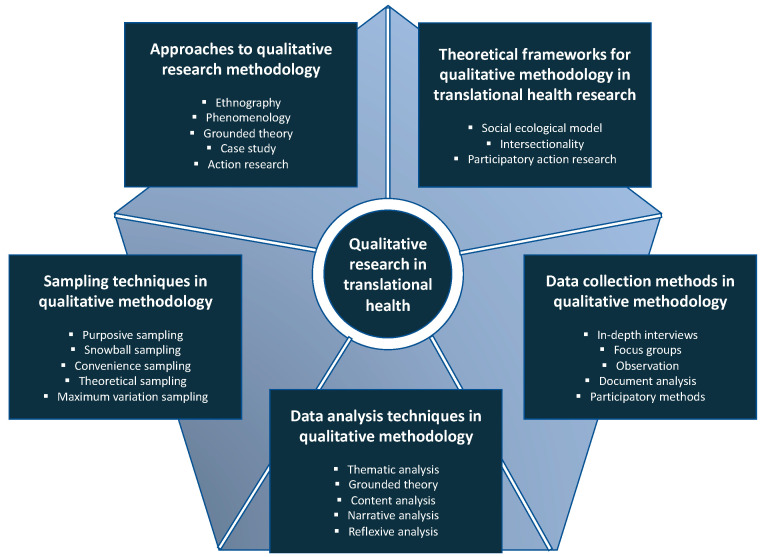
Conducting qualitative research in translational health.

**Figure 2 healthcare-11-02665-f002:**
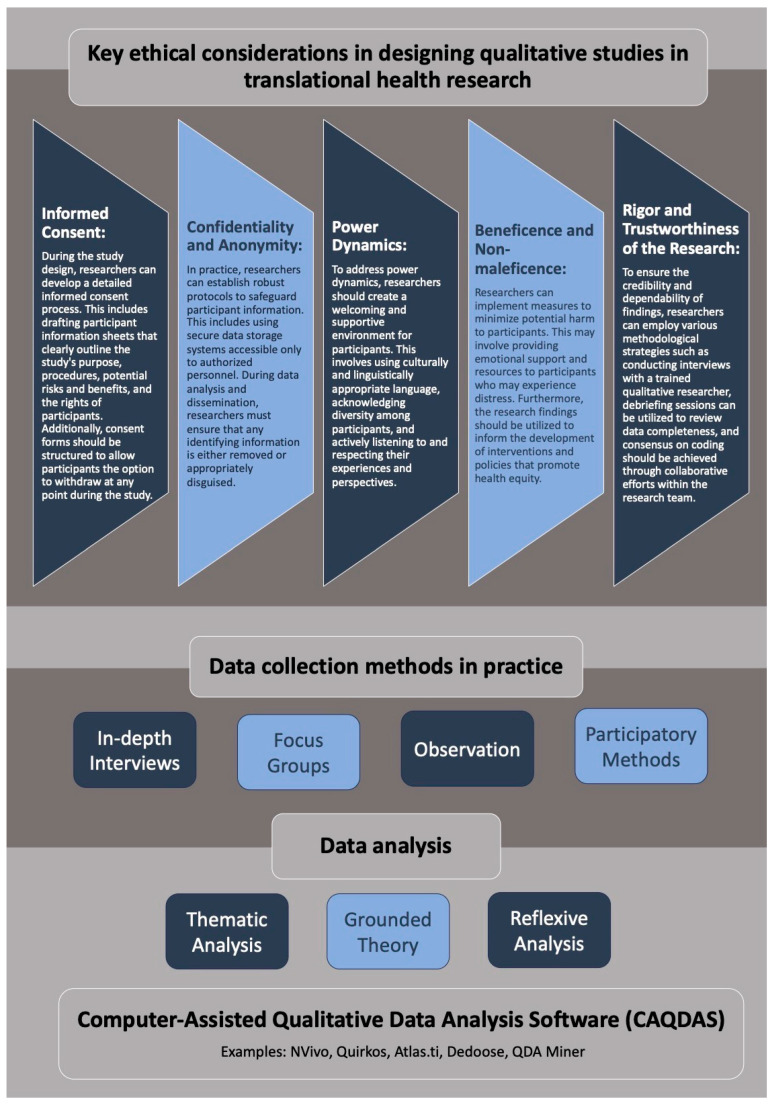
Visual summary of key components of qualitative data collection and analysis in practice in translational health.

**Table 1 healthcare-11-02665-t001:** Qualitative research approaches commonly used in translational health research.

Research Approach	Description	Example Studies
Ethnography	This qualitative approach involves the detailed observation and description of social and cultural practices in a specific setting or community. Ethnography can be useful in translational health research for understanding how cultural and social factors influence health behaviors and healthcare practices [20].	Burnard et al. [21] Hinder and Greenhalgh [22] Barratt et al. [23]
Phenomenology	This approach focuses on exploring the subjective experiences of individuals or groups, with the aim of understanding the meanings and interpretations attached to those experiences. Phenomenology can be helpful in translational health research for uncovering patient perspectives on health and healthcare [24].	Chang et al. [25] Arcadi et al. [26] Hailemariam et al. [27]
Grounded theory	This approach involves developing a theory or explanation of a phenomenon based on data that is systematically collected and analyzed. Grounded theory can be useful in translational health research for generating new insights into the complex and dynamic processes that shape health and healthcare [28].	Sharrock and Happell [29] Ligita et al. [30] Rose and Howard [31]
Case study	This approach involves an in-depth analysis of a specific case or cases, often focusing on the experiences of individuals or groups within that case. Case studies can be useful in translational health research for exploring the unique features of a particular health condition or healthcare system [32].	McDonald et al. [33] Romney et al. [34]
Action research	This approach involves a collaborative approach to research, where researchers work closely with stakeholders to identify problems, develop solutions, and implement changes in real-world settings. Action research can be useful in translational health research for engaging with communities and healthcare providers to improve health outcomes and healthcare [35].	Voigt et al. [36] Livingston et al. [37]

**Table 2 healthcare-11-02665-t002:** Summary of sampling techniques commonly used in qualitative translational health research.

Sampling Technique	Description	Strengths *
Purposive sampling	Purposive sampling is one of the most used sampling techniques in qualitative methodology for translational health research. This technique involves the selection of participants based on specific characteristics that are relevant to the research question, such as age, gender, ethnicity, or health condition. While purposive sampling allows researchers to target participants who are likely to provide rich and relevant data, this approach may limit the generalizability of findings [51,52,53].	Allows researchers to target participants who are likely to provide rich and relevant data. Can be used to achieve diversity within the sample.
Snowball sampling	Snowball sampling is a sampling technique frequently used in qualitative methodology for translational health research, particularly when studying hard-to-reach populations. This technique involves the identification of a small number of participants who meet the inclusion criteria and requesting them to refer other potential participants who meet the same criteria. While snowball sampling allows researchers to access hidden or marginalized populations, it may introduce biases if participants refer others who share similar characteristics [54].	Can be used to access hidden or marginalised populations. Can be effective when studying hard-to-reach populations and sensitive topics.
Convenience sampling	Convenience sampling is a sampling technique that involves the selection of participants who are readily available and accessible for participation in the study, such as patients in a clinic or attendees at a community event [51]. This technique is often used when time and resources are limited [51,53].	Convenient and cost-effective technique. Useful when time and resources are limited.
Theoretical sampling	Theoretical sampling is a sampling technique that is often used in theory-based research. This technique involves the selection of participants based on emerging themes or concepts that are identified during data collection and analysis. Theoretical sampling allows researchers to refine and test their emerging theories. However, this technique requires ongoing data collection and analysis and may result in a smaller sample size [55].	Allows researchers to refine and test their emerging theories. Can be used to achieve theoretical saturation.
Maximum variation sampling	Maximum variation sampling is a technique that involves the selection of participants who vary along several dimensions that are relevant to the research question, such as age, gender, socioeconomic status, or health condition [56]. This technique allows researchers to capture a broad range of perspectives and experiences. However, it is crucial to achieve a balance between diversity and representativeness when using maximum variation sampling [56].	Allows researchers to capture a broad range of perspectives and experiences. Can be used to achieve diversity within the sample.

* The enumerated strengths are not exhaustive and may vary depending on the specific context in which the sampling technique is used.

**Table 3 healthcare-11-02665-t003:** Data analysis framework.

Analysis	Approach of Data Collection and Analysis	Outcome
Thematic Analysis	Data collection and transcription Develop coding framework, read the transcripts, and identify patterns/themes from the data Inductive/deductive or mixed approach	Report major themes with quotes
Grounded	Data collection Form concepts based on a theory/framework Constant comparison and iteration	Validation or development of new theory
Reflexive	Reflection of a researcher’s interaction and positionality in data collection and analysis process Evaluate researcher’s biases and assumptions	Researcher’s reflections/insights on how the experiences, assumptions, bias, and values, influence the study

## Data Availability

Not applicable.

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
