# Peer review of "Qualitative Methodology in Translational Health Research: Current Practices and Future Directions"

_healthcare, 2023, doi:10.3390/healthcare11192665_

Round 1

Reviewer 1 Report

The authors need to check the in-text citation style as it ought to be like this [1], [34] and not (1), (34). Also, if the in-text referencing is serial, then it should be like this [21-23] and not (21,22,23)

L90: emphasizes

L135: Qualitative research…..

L137: reduce the space “but is guided by…”

L195: …interviews. However, …….

None

Author Response

We appreciate your time and expertise in thoroughly reviewing our manuscript. We have revised the manuscript based on all of your valuable comments and provided details of each change in the response letter attached.

Reviewer 2 Report

I would like to thank the handling editor for offering me the opportunity to review the manuscript entitled “Qualitative methodology in translational health research: current practices and future directions” by Rana and colleagues, which is currently under consideration for publication in Healthcare. I would also like to commend the authors for their scholarly work, which presents a review of qualitative research methodology and its application in translational health research. The authors describe various approaches to qualitative research, including ethnography, phenomenology, grounded theory, case study, and action research. Several theoretical frameworks are also examined, such as the social ecological model, intersectionality, and participatory action research, which can provide structure for exploring complex health issues. The review discusses sampling techniques commonly used in qualitative research, including purposive, snowball, convenience, theoretical, and maximum variation sampling. Data collection methods examined include in-depth interviews, focus groups, observation, document analysis, and participatory methods. Data analysis techniques reviewed include thematic analysis, grounded theory, content analysis, narrative analysis, and reflexive analysis.

The authors emphasize the value of qualitative methodology in translational health research for understanding social, cultural, and contextual factors influencing health and healthcare. Qualitative approaches can complement quantitative findings within a mixed methods design. The review highlights ethical considerations in conducting qualitative research on sensitive health topics. Challenges and limitations are discussed, including subjectivity, small sample sizes, validity, and resource constraints. Future directions are suggested, such as leveraging digital technology, incorporating mixed methods, and using participatory approaches to promote health equity. Overall, the review demonstrates the applicability of qualitative methodology in translational health research and provides guidance for researchers in this interdisciplinary field.

This comprehensive review examines the application of qualitative methodology in translational health research and makes a valuable contribution to the literature. The manuscript appears scientifically and technically sound, outlining key qualitative approaches, sampling techniques, data collection methods, and analysis strategies. The authors thoughtfully consider ethical issues involved in qualitative health research, such as informed consent, confidentiality, power imbalances, and potential participant distress.

The review has several strengths. It provides a detailed overview of fundamental qualitative methods and how they can be utilized to explore complex health issues and complement quantitative data. The authors highlight the ability of qualitative approaches to capture unique insights into social, cultural, and contextual factors shaping health and healthcare. The examination of theoretical frameworks offers a structured lens for interpreting qualitative findings.

The manuscript occupies an important niche in the literature at the intersection of qualitative methodology and translational health research. While existing reviews have focused on qualitative methods in general healthcare research, this review specifically targets the emerging interdisciplinary field of translational health. It makes a novel contribution by emphasizing the value of qualitative data in bridging gaps between basic science, preventative research, and clinical practice.

The authors comprehensively synthesize prior literature and offer additional original perspectives. The review is forward-looking in its coverage of ethical considerations, limitations, and future directions like participatory methods and mixed methods approaches. Overall, the manuscript is scholarly, well-referenced, and clearly situated within the current literature. It provides helpful guidance for employing qualitative methodology in translational health research and could make a meaningful impact in this rapidly evolving field.

While the manuscript provides valuable insights, there are a few areas that could be refined to further augment the quality and impact of the work. Here are some respectful suggestions to potentially improve the manuscript:

·         The abstract provides a concise overview of the scope and content of the review. However, the authors could consider clearly outlining the key objectives, research questions, or knowledge gaps being addressed. This would signal to readers the focused aims of this review and enhance understanding of its unique contribution to the literature.

·         The introduction effectively establishes the context and significance of qualitative methods in translational health research. Nonetheless, the authors could cite any recent statistics or trends that demonstrate the growth of this research approach to further accentuate its importance.

·         The introduction focuses on the utility of qualitative data but could also note its limitations. This would provide a more balanced presentation of the subject matter.

·         When describing ethical issues, the authors could include a section examining strategies for establishing trustworthiness and rigor in qualitative research design. A focus on optimizing trustworthiness would strengthen this methodological review by underscoring the authors' commitment to scientific quality, enhancing the credibility and applicability of qualitative findings, and building a conceptual foundation for later sections.

·         Furthermore, when describing ethical considerations, the authors could provide concrete examples of how these principles can be upheld in practice during study design and implementation. This would demonstrate how these theoretical concepts can be practically applied, enhancing understanding for readers.

·         For data analysis methods, the authors could provide a visual summary depicting their relationship to different qualitative approaches. This would break up a text-heavy methods section and cater to readers who absorb visual information more readily. A schematic diagram would complement the detailed explanation provided in the text and serve as a handy reference point.

·         The limitations and challenges are well described. To balance these, the authors could elaborate on techniques, strategies, or guidelines for effectively addressing these challenges in real-world research.

·         When presenting future directions, the authors could rank or prioritize which innovations hold the most promise or potential for the field. This would help readers understand which advancements the authors view as most impactful for progressing the field of qualitative methodology in translational health research.

·         The authors could compare the strengths and weaknesses of leveraging digital technology versus traditional in-person methods for data collection and engagement. This could provide a balanced analysis to help readers understand the appropriate applications of each approach.

·         The authors could provide exemplar studies that demonstrate the meaningful impact qualitative findings can have on health policy or practice. This could demonstrate the real-world influence qualitative findings can have, enhancing the perceived value of this methodology.

·         The conclusion succinctly summarizes the review. Nevertheless, the authors could end with a call for more research, education, or policy changes needed to support rigorous qualitative translational health research and practice.

In conclusion, I would like to reiterate my appreciation to both the editor and the authors for the opportunity to review this intriguing and informative manuscript. I trust that my suggestions will help enhance the quality, clarity, and impact of this extensive review for journal readers and translational health researchers. I look forward to seeing the revised version of the manuscript and wish the authors success in their ongoing research endeavours.

Author Response

(The authors gave the same response as above.)

Reviewer 3 Report

Manuscript reviewing the use of qualitative research in translational research, with a brief summary of the use of this methodology. It is a good introduction to the subject for researchers who are not experts in this topic, but it does not offer enough depth to start doing it. I consider it an interesting and useful manuscript. However, the analysis of data and instruments is a complex issue in qualitative research, so I suggest adding analysis mechanisms, including computer programs, used in each strategy. In this way, it will guide the readers on the strategies and some recommended way to analyze their results. Just add an orientation, so that those interested can investigate these analysis strategies in detail in other specialized texts.

Author Response

(The authors gave the same response as above.)

Round 2

Reviewer 2 Report

I want to express my appreciation for the attention and consideration you have devoted to my suggested revisions for your manuscript. It is evident that a significant amount of effort and thought has been directed towards the refining of your work, integrating the feedback provided during the peer review process. The resulting modifications demonstrate a thorough and thoughtful approach, and significantly enhance the rigor and overall quality of your manuscript. I look forward to witnessing the impact your research will undoubtedly have on the academic community.

Author Response

Thank you once again for taking the time to review our paper. We are truly grateful for your valuable feedback throughout this process.

Reviewer 3 Report

The manuscript has been improved. I recommend publishing it

Author Response

(The authors gave the same response as above.)
